# Small Vessel Disease: Ancient Description, Novel Biomarkers

**DOI:** 10.3390/ijms23073508

**Published:** 2022-03-23

**Authors:** Rita Moretti, Paola Caruso

**Affiliations:** Neurology Clinic, Department of Medical, Surgical and Health Sciences, University of Trieste, 34127 Trieste, Italy; paolacaruso83@gmail.com

**Keywords:** small vessel disease, vascular damage, blood–brain barrier damage, reactive oxygen species, endothelial dysfunction, metalloproteinases

## Abstract

Small vessel disease (SVD) is one of the most frequent pathological conditions which lead to dementia. Biochemical and neuroimaging might help correctly identify the clinical diagnosis of this relevant brain disease. The microvascular alterations which underlie SVD have common origins, similar cognitive outcomes, and common vascular risk factors. Nevertheless, the arteriolosclerosis process, which underlines SVD development, is based on different mechanisms, not all completely understood, which start from a chronic hypoperfusion state and pass through a chronic brain inflammatory condition, inducing a significant endothelium activation and a consequent tissue remodeling action. In a recent review, we focused on the pathophysiology of SVD, which is complex, involving genetic conditions and different co-morbidities (i.e., diabetes, chronic hypoxia condition, and obesity). Currently, many points still remain unclear and discordant. In this paper, we wanted to focus on new biomarkers, which can be the expression of the endothelial dysfunction, or of the oxidative damage, which could be employed as markers of disease progression or for future targets of therapies. Therefore, we described the altered response to the endothelium-derived nitric oxide-vasodilators (ENOV), prostacyclin, C-reactive proteins, and endothelium-derived hyperpolarizing factors (EDHF). At the same time, due to the concomitant endothelial activation and chronic neuroinflammatory status, we described hypoxia-endothelial-related markers, such as HIF 1 alpha, VEGFR2, and neuroglobin, and MMPs. We also described blood–brain barrier disruption biomarkers and imaging techniques, which can also describe perivascular spaces enlargement and dysfunction. More studies should be necessary, in order to implement these results and give them a clinical benefit.

## 1. Introduction

Small vessel disease (SVD) (also called cerebral small vessel disease, cSVD) relies on the deep brain’s small vessels alterations. *Small vessels* are univocally defined as small penetrating arteries, capillaries, and small veins. cSVD is strongly related to a chronic hypoperfusion condition, which predisposes the entire brain to hemorrhagic events, white (confluent or not) matter alterations, and lacunar events. SVD is the most important and common cause of all the vascular forms of dementia (up to 45%), but as previously underlined, it predisposes to a higher risk of vascular strokes (25–30% of cases) and 25–35% of all the lacunar events [1,2]. The other crucial common characteristic of SVD is that its pathological consequence could be represented by a silent lesion progression, which has its clinical confirmation in dramatic radiological imaging, without apparent, evident acute events. Thus, SVD is a clinical condition whose principal stigma is that the lesions may progress over time, for imprecise rules, and above all, with or without clinical consequences, in relationship with the extension and the confluency of the white matter alterations [3,4,5].

Generally, SVD clinical signs are concomitant psychological and behavioral sequelae, summarized by an essential executive function disruption and standard neuropsychological features (apathy and vascular depression) [1,2,3,4].

In sporadic cerebral SVD, aging, diabetes, chronic hypoxia, and hypertension are the most recognized clinical risk factors. Still, different hereditary forms of cerebral SVD have also been described [6]. Small arterioles show significant disruptions in both cases, easily described as arteriolosclerosis, lipohyalinosis, and severe endothelial disruption. Its principal consequence is a strong invalidation of the neurovascular coupling mechanisms and vessel tone dysregulation [7,8], and even venules are interested in the ongoing process [9]. With the arteriolosclerotic process, SVD is characterized by a substantial increment in cerebral amyloid angiopathy (CAA). This condition, which has been traditionally related to Alzheimer’s disease, is, on the contrary, quite frequent in the normal aging process, and it is dramatically evident in the SVD process. It is related to a consistent deposition of amyloid b-peptide (Ab) in the walls of the small arterioles, and it increments the consequences of altered neurovascular coupling in small parenchymal and leptomeningeal arterioles [9,10,11]. 

The principal consequence of arteriolosclerosis is the chronic hypoperfusive state, which induces a perpetual neuro-inflammation state, and gives rise to an essential endothelial activation. These conditions induce an overwhelming alteration of the oxidative response, which potentiates the basal inflammation status of the deep brain structure, expanding through different neural networks, principally the basal-forebrain ones [12]. 

In a recent review [12], we focused on the contribution of the complex and multifaceted “vascular damage” in developing small vessel dementia, starting from small vessel disease condition. We have written that the SVD is “an ongoing process, which begins with altered microvessels and pial arteries and ends in subcortical dementia; CBF regional selective decrease seems to be one of the critical factors for the progression from small vessel disease to small vessel disease-related dementia, together with proved altered response to inflammation, and oxidative stress” [12]. 

Neuroimaging is the main helpful diagnostic instrument for managing brain SVD. Therefore, the main findings in SVD are subcortical infarcts, lacunes, white matter hyperintensities (WMHs), prominent perivascular spaces (PVS), and cerebral microbleeds (CMBs) [13]. T2 or FLAIR MRI reports indicate confluent and symmetrical white matter hyperintensities [14,15,16,17,18,19,20,21,22,23] into the frontal and prefrontal-thalamus-basal forebrain networks [24,25,26,27,28,29]. Many instruments have been implemented to relate the number of lacunes, the extension and the amount of surface of white matter hyperintensities, and their relation to the subsequent cognitive and behavioral impairment [30,31,32]. The confluence between clinical, neuropsychological, and neuroimaging findings helps to converge for a correct diagnosis of the Vascular cognitive impairment, as stated in NINDS-AIREN criteria [33,34,35] and the DSM-V R (Fifth Edition-revised) [36,37,38,39]. VCI refers to an ample spectrum of vascular brain pathologies that contribute to cognitive impairment, ranging from mild and subjective cognitive decline to overt dementia [40,41].

The state of the art in the VCI field is an ongoing definition [42], and many terms have been employed, such as the descriptive ones of vascular cognitive disorder, subcortical vascular dementia, and mild and major vascular neurocognitive disorders (Diagnostic and Statistical Manual of Mental Disorders, Fifth Edition (DSM-5) [36,37,38,39,43]. Others include vascular cognitive disorder (VCD), while subcortical VAD (sVAD) has been employed to define a circumscribed syndrome, related to small vessel disease [44,45,46]. However, it is well defined that the difference between VaD subtypes may depend on the anatomical distribution of the vascular insults [47]. Usually, small artery disease is more often associated with subcortical VaD than with cortical and cortical-subcortical VaD [47]. Executive dysfunctions and behavioral disorders (apathy and vascular depression, etc.) are the commonest findings [48]. 

Very recently, excellent studies emerged on the potential role of transcranial doppler findings in patients with white matter lesions, as possible markers for developing vascular dementia. A significant example is the demonstration that patients with white matter lesions, but without any other sign of cognitive impairment, showed a hemodynamic patter of cerebral hypoperfusion and enhanced vascular resistance, as a distinctive marker of a possible predictor factor of developing dementia [49]. Another significant result is the one obtained by a study using a transcranial Doppler, in which patients with ≥80% unilateral internal carotid artery stenosis with no history of stroke were recruited [50]; this study demonstrated that cognitive impairment correlated linearly with lower flow in the hemisphere fed by the occluded internal carotid artery, but only below a threshold of MFV = 45 cm/s. [50] 

Finally, transcranial doppler studies have shown that it could delineate a profile of low perfusion and high vascular resistance in patients with a defined diagnosis of vascular depression [49].

In this paper, we wanted to address the contribution of the chronic inflammatory brain condition due to SVD and, starting from this situation, verify the possibility of finding new biomarkers of endothelial dysfunction, inflammation, and oxidative damage, which could be possible future targets of focused therapies.

## 2. Possible and Proved New Markers of Blood–Brain Barrier Leakage, Perivascular Enlargements, and Mitochondrial Alterations

SVD has the small vessels (pial and the small penetrating) and white matter as a significant definite target. Nevertheless, growing attention has been dedicated to disrupting perivascular spaces, astrocytic end-feet, capillaries, and veins. As a final point, the blood–brain barrier (BBB) has been addressed as another potential target of the intrigued mechanisms that underlie the small vessel brain pathology complex. BBB is not only a solid defensive barrier but acts as an active and specific player of active selection crossover, possessing cell-cell signaling with the end-feet of astrocytes and disclosure a potential role of maintaining efflux pumps [51,52,53,54,55]. Thus, the disruption of the BBB is proportionately increased by normal aging but progresses as a hallmark in different pathologies, i.e., multiple sclerosis or in primary inflammatory disease. Nevertheless, it is an expression of white matter inflammation, even due to chronic hypoperfusion, such as the one which occurs in small vessel disease [SVD], accomplishing the progression and the extension of the white matter sufferance, named as white matter hyperintensities (WMH) [56,57,58,59,60,61,62], the confluency of which is synonymous with SVD progression, leading to subcortical vascular dementia (sVAD) [12,62]. In AD-prone patients, BBB disruption has been signaled even in hippocampal degeneration, which occurs after a major stroke [61,62]. 

A dynamic contrast-enhanced MRI (DCE-MRI) [63] has been employed for in-vivo quantification of the pathological passage of plasma through BBB [64,65]. Moreover, apart from the BBB leakage, the possibility of estimating the vascular permeability-surface area product (PS) and the plasma volume fraction (VP) in a given region of interest has also been described [66,67]. The model suggested that PS increased with WMH severity, aging, and other vascular risk factors, and at the same time, a lower blood vP [65]. The most promising in-vivo demonstration is that BBB integrity is compromised in more severe WMH, even beyond visible lesions [63] (Insert Figure 1).

Even if we know that BBB is disrupted in SVD, we do not know the reasons for BBB leakage in this condition. The most disputed involvement is one of the pericytes. Pericytes are capillary mural cells that stabilize newly formed vessels and induce repair. When a pericyte-deficient adult mouse model has been employed [68], different transcriptional changes in brain endothelial cells have been mapped due to a defective pericyte contact at a single-cell level. In that conformation, endothelial cells, deprived of pericyte contacts, seem to exhibit a “venous-shifted molecular pattern,” and therefore lack any capillary specialization, and upregulate proteins which are typically expressed during developmental stages, such as the Fibroblast Growth Factor Binding Protein (*Fgfbp1*), or those expressed during pathological angiogenesis, such as Angiopoietin 2 (*Angpt2*). These aspects permit a possible cell proliferation, with a very flawed arteriolar BBB regulation system, and reduction of the angiogenesis process [68]. *Fgfbp1* and *Angpt2* levels could probably be crucial markers of BBB leakage during SVD. More studies will be necessary to prove that.

Perivascular spaces (PVS) have gained an essential role in SVD pathogenesis; they are no longer considered as virtual empty spaces, but as the most efficacious catabolites clearance system [12]; they are resident sites of perivascular macrophages, pial cells, mast cells, nerve fibers, and collagen fibers [69]. PVS are virtual spaces intimately connected to deep arterioles [70]. Even in these conditions, they act as a lymphatic net, defined as a glymphatic-perivascular territory [71]. 

Their malfunction, the hallmarks of which are the combined enlargement and widening, is the principal responsibility for perivascular accumulation of catabolites and toxic substances, which is determinant for enhancing ongoing neural damage until starvation [72,73]. The perivascular debris accumulation, together with the BBB leakage, potentiates and accelerates the perivascular inflammation, strongly favored by the stagnation-induced process and by medical conditions which influence it, such as hypertension and diabetes [74,75,76,77,78]. PVS enlargement is responsible for an altered cerebrovascular reactivity (CVR) [12], due to the extension of the constant inflammatory response [41] present as a constant marker in SVD, due to the chronic hypoperfusion state. The PVS is never an isolated situation, but it is accompanied by an altered BBB disruption and a significant perivascular inflammation [75,79,80,81,82]. More recently, new actors contribute with BBB leakage and PVS enlargement to help the progress of SVD [83,84,85,86], such as the oligodendrocyte precursor cells (OPCs), which generally help BBB stabilization [86,87] and the astrocytes, which exert their fundamental role as regulating the signal of neuro-vascular coupling [12]. Oligodendrocytes are the first victims of chronic models of chronic cerebral hypoperfusion (CCH), together with the precocious sufferance of the perineural space [88,89,90], and with a hyper-activation of microglia, firstly in the hippocampus [91,92], then in the thalamus, up to in the cortical neuronal population [93]. Secondary to oligodendrocytes, astrocyte death occurs in proportion to the chronic ischemia condition’s length and severity [94,95], due to the ongoing modifications of general and neuronal metabolic requests. Their death is a consequence of chronic hypoxia, but it worsens neuronal death due to a lack of functions, regulating the neurovascular coupling signal [96]. The process by which this occurs is that during the entire process of chronic ischemia, microglia retract its branches, with a consequent reduction of the length and strength of the microglial ramification, with a concomitant degeneration of the soma [97]. The frontal activation of microglia occurs in a two-step pattern: at the beginning, M1 activation upregulates TNF alpha, Il-23, IL-1beta, and Il12 production, which attack neurons, and directly contribute to their injury; only after M2 activation occurs can the reparation process can begin [98]. In the SVD, due to the chronic hypoxia-hypoperfusion condition [12], the passage through M1 towards M2 activation does not occur [98]. In SVD, there is a substantial augmentation of M1 activation, together with a heavy reduction of M2 promotion [99,100]. The brisk oligodendrocyte degeneration, associated with M1 activation, increases calcium currents and induces a severe apoptosis process. The calcium increases, and the severe apoptosis is accompanied by an augmentation of caspase-3 RNA and matrix-metalloprotease 2 (MMP-2) [101]. At the beginning of the SVD process, these markers reflect the temptation reparation process induced by a standard M1/M2 passage, as described above. Nevertheless, until the chronic inflammatory condition occurs in SVD ongoing development, there is an alteration of the M1/M2 passage, with a predominant M1 event; therefore, in SVD patients’ cerebrospinal fluid (CSF), there is a constant growth of oligodendrocyte-derived myelin sheath-like myelin lipid sulfatide (ODMSMS) and myelin essential protein (MBP) due to the massive oligodendrocytes death [102,103,104,105]. For similar reasons, markers of axonal damage, i.e., neurofilament light chain (NFL), together with CSF α-1 antitrypsin, tissue inhibitor of metalloproteinase-1 (TIMP-1), plasminogen activator inhibitor-1 (PAI-1), and apolipoprotein H (ApoH) have been found to increase very early in the CSF in SVD [106,107,108]. Finally, due to the BBB leakage, ultrastructural studies find that in older animals as well as in those affected by SVD, there are severe alterations of the capillary basement membrane of the deeper arterioles, inside the white matter, filling plasma proteins into vascular bagging and collagen deposition inside PVS, in a phenomenon described as microvascular fibrosis [55,98,109]. Many studies have testified that microvascular fibrosis and BBB splitting have a higher CSF/serum albumin (SA) ratio in patients with SVD [109]. Matrix remodeling pathway (TIMP-1 and matrix metalloproteinases) as an expression of endothelium disruption in SVD has been described [109] (Insert Table 1 here). 

## 3. Markers of Endothelial Dysfunction

As previously described [12], there is a global endothelial altered function in SVD [110,111], which could be synthesized in an alteration of normal endothelial response to endothelium-derived nitric oxide-vasodilators (ENOV) [112], prostacyclin [113], C-reactive proteins [114], and endothelium-derived hyperpolarizing factors (EDHF) [115]. 

NO is rapidly removed in SVD for the mitochondrial alterations, with a consequent anti-oxidative response and consumed by peroxynitrite (O2 anions plus NO) [116]. However, it can also be reduced in its production, as it occurs in normal aging [117], in an accelerated way, in SVD, with a consistent down-regulation of endothelial NO synthase (eNOS). Moreover, in SVD, there is an evident dysfunction of the Rho-associated protein kinase (ROCK) [118] and the related ERM proteins (ezrin, radixin, and moesin), fundamental for barrier properties’ integrity [118,119,120] and their induction of the downregulation of the vascular endothelium cadherins (VE-cadherins) [121]. 

In diabetes, where SVD is a constant presentation form with a crucial endothelial hyper-permeability, a concomitant increase in arteriolar deposition of advanced glycation end products has been observed, which helps and maintains the increase in endothelial permeability through Rho activation and an upregulation of the vascular endothelial growth factor (VEGF) [122,123]. 

The superimposition of BBB disruption, endothelial dysfunction, and microvascular fibrosis causes a substantial permeability alteration, with albumin extravasation; the increased CSF/plasma albumin ration is a proven witness of a severe progression of confluency of white matter lesions in SVD [124,125,126,127], together with albuminuria (even if not well-accepted) [128,129,130,131]. 

Other important markers of endothelial altered activation [12,132,133,134] in SVD are intercellular adhesion molecule-1 (ICAM-1), which has been considered as a generic expression of white matter progression [95], soluble thrombomodulin (sTM), interleukin-6 (IL-6), plasminogen activator inhibitor-1 (PAI-1), and von Willebrand factor [129,130,131,132,133,134]. Others, such as HIF 1 alpha, VEGFR2, and neuroglobin, are more evident when the confluency of different WMH becomes constant in different models [135,136]. 

## 4. Markers of Oxidative Damages in SVD

Reactive oxygen species (ROS) is an umbrella term for many ordinary derivatives of molecular oxygen, and their accumulation leads to a complex phenomenon called oxidative distress. There are two species, hydrogen peroxide (H_2_O_2_) and the superoxide anion radical (O_2_^−^), which are key redox signaling agents generated under the control of growth factors and cytokines by more than 40 enzymes, prominently including nicotinamide adenine dinucleotide phosphate (NADPH) oxidases [12] and the mitochondrial electron transport chain [126]. When mitochondrial cells usually function, the active process of oxidative phosphorylation converts oxygen to superoxide by oxidase enzymes, and superoxide can be transformed by superoxide dismutase (SOD) or to non-radical hydrogen peroxide [126,136,137], i.e., from glutathione peroxidase (Gpx), or when catalase enzymatically metabolizes hydrogen peroxide to water and oxygen [136]. 

Chronic cerebral conditions of constant hypoxia are the principal inductors of the uncontrolled production of ROS [138,139]. 

NADPH oxidase activity and mitochondrial are significantly higher in cerebral arteries when compared with systemic arteries in blood vessels from healthy animals (mouse, rat, pig, and rabbit) [140,141]. Thus, brain vessels are one of the most prominent productions of ROS, suggesting that there could be fundamental ROS-dependent signaling in cerebral arteries, which might be indispensable for vasoactive regulation properties. 

Thus, the accumulation of ROS species, associated with mitochondrial dysfunction, BBB disruption, and chronic inflammatory status are three conditions in SVD and are proportionate to WMH extension. They lead to an altered endothelial further altered activation, which is reflected in a decoupling of the neurovascular coupling system, with significant sub-cortical and cortical signal alteration, with consequent reflex in oligodendrocytes astrocytes and finally to neurons [12,142]. An active role of flow-dependent responses in rat cerebral arteries has been recently demonstrated in vivo, directly exerted by the NADPH-oxidase reactions [143]. Specifically, Nox_2_-NADPH oxidase dysfunction is related to the propagation of the ischemic brain injury, derived by the occlusion of larger pial arteries; Nox_2_/NOx_2_ knock-out mice, in the same condition, show the minor extension of brain injury after an ischemic infarct [144]. 

The induced alterations of mitochondrial DNA by ROS attacks and chronic ischemic conditions are some of the most critical contributors to neuronal aging and degeneration, either considering oxidative damage as a promoter or as a consequence of it [145,146,147]. 

The decline of mitochondrial functioning has been largely implicated in the aging process and is characterized by a reduced density of mitochondria and reduced mitogenesis [148,149,150,151,152]. Such changes, which originate as replication errors, accumulate in postmitotic tissues during aging, leading to increased proportions of impaired mitochondria [152]. In the aging brain, there has been a sufficient demonstration of impairment of synaptic mitochondria leading to impaired neurotransmission and cognitive failure [149,150,151,152,153,154,155]. Precocious forms of small vessel disease, leading to vascular dementia, have been described in specific mitochondrial point mutation [156]. Other mitochondrial mutation phenotypes have been described as pure brain involvement, including fluctuating encephalopathy, seizures, dementia, migraine, stroke-like episodes, ataxia, and spasticity [149,153,154,155]. Growing attention should be paid to mitochondrial DNA mutations for brain pathologies, in order to gain more robust data on their possible relevance, and their correlation with postmortem neuropathologic features, to advance our understanding [156,157,158,159,160,161].

Oxidative stress potentiates the disorders of the endothelium-dependent NO signaling [162,163]. Uncoupling endothelial NO synthase (eNOS) (i.e., in relation with lower levels of tetrahydrobiopterin) switches the production of NO to that of superoxide, causing an overwhelming potentiation of ROS production, accelerating the oxidative stress, lowering the NO anti-inflammatory properties [164,165], and reducing NO modulation of Rho-kinase activity, inhibiting vascular tone control [166]. Rho-kinase, as a counterpart, influences mRNA-stability of eNOS [167].

The induction of oxidative stress is one of the most important promoters of pathological angiogenesis, by lipid oxygenation, thickening the blood vessel walls [168,169]. Moreover, the ApoE4 allele and the AD process seem to be involved in promoting vascular alterations independently of other recognized factors, i.e., age, diabetes, hypertension, and obesity, etc. However, it is supposed to worsen the confluency of WMH, probably somehow linked to ROS augmentation, without any other positive data [170,171,172].

## 5. Inflammation and SVD

As above written, neuroinflammation is a common finding in SVD models; it is tightly related to chronic hypoperfusion condition and defined as located hypoxia condition, the common finding of SVD. The pivotal role of neuroinflammation in SVD could accelerate the lipid peroxidation precipitation of the redox system and promote a more robust activation of M1 than M2 [173].

It has been demonstrated that NO-related metabolite, citrulline, and dimethylarginine (DMA) concentrations were significantly higher in patients with strategic infarcts [174]. Arginine depletion was an independent predictor of VaD [174]. S100B (calcium-binding protein B) is a protein that stimulates the expression of pro-inflammatory cytokines. SomIt has been described to have a significant correlation between S100B/asymmetric dimethylarginine levels and cognitive decline in patients with leukoaraiosis [175,176].

Homocysteine could be a potential marker of neuroinflammation inside SVD, promoting the increase in TNF-alpha and IL1-beta, upregulating the transcriptional fibroblast growth factor-2, IL-6, and IL-8, [177,178], and enhancing the VEGF/ERK1/2 signaling pathway [179,180], which can be seen frequently in the atherosclerosis process. Homocysteine is directly linked to the B-inflammatory pathway through a direct upregulation of pyruvate kinase muscle isoenzyme 2 (PKM-2), B-mediated, which mainly promotes the inflammatory basis of atherosclerosis cascade [181,182]. 

Homocysteine accumulation promotes an increase in the endoplasmic reticulum (ER) stress, upregulating metalloproteinases-9 (MMP-9), and inducing apoptosis [183]. Definitively, the accumulation of homocysteine in animal models enhanced the expression of the AGEs or vascular cell adhesion molecule [184] and MMP-9 [185]. The inflammation cascade could be mediated by the effects on smooth muscle cells rather than on the endothelium alterations [186,187]. 

Chronic inflammation and oxidative stress have been suggested as concurrent mechanisms of SVD. A possible link between accumulation products (i.e., homocysteine) and other markers could be the circulating metalloproteinases (MMPs) and the tissue inhibitors of metalloproteinases (TIMPs) [188]. 

MMPs, some of the Ca^2+^-Zn endopeptidase, have been described as having six different properties: collagenase, gelatinases, stromelysins, matrilysin, membrane-specific metalloproteinases, and no other specified. Their specific role inside the brain is complex and multifaceted; it begins with the neuronal networks remodeling throughout the integrity of the BBB [189]. MMP remains inside the brain, probably in inactivate form, and is active only under special conditions, such as chronic hypoperfusion or chronic inflammatory status. The significant components are MMP2 and MMP14, which are present specifically inside astrocytes, whereas microglia present the MMP-3 and the MMP9, which, by definition, are called inflammatory metalloproteases. Their expression is more severe in acute damage and gradually decreases in the reparation phases. They can be found near the damaged areas and in the propinquity vessel-related areas [190].

There are four possible mechanisms which have been related to MMP involvement in SVD and, in general, in the neuroinflammation process. The most obvious and well-studied MMP directly activates signaling cytokines, cell-receptors, and adhesion molecules. There are essential works that testify that, even directly, MT4-MMP upregulates a TNF-alpha convertase, and is able to activate TNF-alpha, in its soluble and active [191,192]. Secondly, there are many pieces of evidence in different clinical cases (neurological bacterial infection and PD, etc.), in which there is a direct activation, probably mediated by lipopolysaccharides, calcium currents and other apoptotic signals, and alpha-synuclein deposits, which activate MMP-3 into the interstitial brain fluid, and there, it triggers M1 activation, with a consequent (and above-described) M1 activation [193,194].

Thirdly, MMP seems to be tightly involved in the so-called Fas-FasL system. This system has been known as an inducer of extrinsic cell death responsible for cell-mediated cytotoxicity and peripheral immune regulation. MMP might improve the FAS system, probably through an intrinsic possibility of modulating chloride channel activity, inducing and promoting glutamate excitotoxicity currents, or altering the interactions between neuronal cells and extracellular matrix compounds [195].

Finally, the MMPs participate in many digestive processes at the BBB, particularly the tight junctions and the basement membrane. It has been proposed that MMPs digest tight junctions and basement membrane proteins, thus contributing to BBB leakage [196]. The increased activity of MMP, tightly associated with a higher permeability at the BBB, has been demonstrated in vivo during the reperfusion process through an increase in MMP-2 and MMP-9 mRNA activity [197]. The induction of BBB leakage has, as an indirect effect, an increment in the vasogenic edema, inside the WM, with a drastic increment in vascular demyelination process (Insert Figure 2).

These data have been evoked in animal models and rare human models, and there is a substantial lack of information, i.e., on the possible relationship between MMP levels and extension and repairing of stroke lesions [198,199]. 

Nevertheless, some interesting points shed some light on the topic: increased confluency of WMH could be related to higher levels of TIMP-4, after three months of a primary stroke [199].

In a recent study, Arba et al. showed that increasing the grade of SVD sustains higher levels of TIMP-4 and supports the involvement of TIMP-4 in the pathologic process of SVD; they studied a population of an ischemic stroke patient, reporting that brain atrophy was associated with baseline TIMP-4 levels and leukoaraiosis was associated with 90-day TIMP-4 levels. A global SVD score, expressed as a combined product of leukoaraiosis, lacunes, and brain atrophy, was associated with TIMP-4 levels at 90 days with a dose-response effect [199]. 

Increased levels of MMP have been associated with severe white matter alterations and a cognitive profile that resembles sVAD [200,201]. In particular, a positive relationship between MMP2 lower levels has been found, together with an increase in albumin index in CSF of SVD patients, as above written [202,203,204].

Due to lipohyalinosis substitution of smooth muscle cells in arterioles (as described in 12), there is an inverse correlation between TIMP-4 elevated levels (only in animal models) and reduction of lipohyalinosis and collagen bagging through an undescribed and uncertain mechanism [205,206,207].

All these aspects accounted for, MMPs and tissue inhibitors of metalloproteinases-1 (TIMP-1) could be promising SVD biomarkers [208,209]. 

## 6. Potential Future Therapies Approach

Different approaches can be employed to offer potential treatment for VCI, at the moment these are only symptomatic; many data have been obtained from cholinesterase inhibitors and memantine [210]. 

Potential treatment strategies for brain SVD might include those that target antioxidant effects for the endothelium of small cerebral vessels and the BBB. Due to the major decrease in NO bioavailability in SVD, NO donors could help release the functioning endothelium of small vessel disease, limited by their susceptibility to tolerance development. The apparent strategy, in the same manner as the administration of potent antioxidants such as Vitamins C and E, has shown to be beneficial for vascular function in several experimental and small clinical trials [211]. 

Disappointingly, the results of large clinical trials of antioxidant supplementation have largely failed to show any benefit. The ROS scavenger tempol is cell-permeable and has been used in experimental studies, as well as edaravone (O_2_-scavenger). The ROS scavenger tempol is cell-permeable and has been used in experimental studies, as well as edaravone (O_2_-scavenger). Problems derived from NADPH oxidase activity, particularly its primary contributor, Nox_2_. It can be argued that prolonged selective therapies could help prevent brain SVD but invariably lead to an immunosuppression condition and many other side effects derived by other different Nox oxidases [137,145,212]. 

Notably, three of the most influential and frequently prescribed classes of drugs for the treatment of vascular risk factors, which have been shown to inhibit NADPH oxidases, reducing oxidative stress, are the Angiotensin-converting-enzyme inhibitors (ACE inhibitors), Angiotensin II receptor type 1 (AT 1) antagonists, and the statins [3,4]. There are no impressive studies on these drugs as primary NADPH inhibitors, rather than their well-known function per se. 

Many other trials have been conducted and are still ongoing [213].

Phenolic acids (or phenolcarboxylic acids) are aromatic acid compounds containing a phenolic ring and an organic carboxylic acid function [214]. Among the most studied molecules belonging to this group, caffeic, chlorogenic, o-coumaric, p-coumaric, m-coumaric, ferulic, and cinnamic acids are the most commonly consumed in the human diet, being contained in coffee [215], together with gallic, p-hydroxybenzoic, vanillic, syringic, and protocatechuic acids. They can be found in bran, grain brown rice, olive oil, tea, cherries, plums, gooseberries, and red wine [216]. These substances have been studied among middle-aged adults, showing a benefit of their intake in different cognitive domains [217,218,219,220,221,222,223,224].

In the same way, rosmarinic acid induced a promotion of oxidative stress response and a reduced lipid-peroxidation [225], but also reduced the gene expression of inducible nitric oxide synthase [225,226], and promoted neuroprotection, reducing matrix metallopeptidase 2 (MMP2), and IL-1 beta [225,226,227]. Myrtenal has been recently employed as a multi-property substance (anti-inflammatory and anti-oxidant) [228] but results are only promising.

Apart from physical aerobic activity and avoiding vascular risk factors (smoking, high quantities of carbohydrates, and alcohol consumption, etc.), even external stimuli have been applied in studies; in order to implement cognitive abilities on vascular deterioration, transcranial magnetic stimulation has been studied, which it is still under debate, because its activity on the dorsal striatum with the consequential increase in dopamine release may contribute to the clinical and neurophysiological outcome in vascular depression and vascular cognitive impairment [229] (Insert Figure 1).

## 7. Conclusions

In the last few decades, the concept of vascular contributions to cognitive impairment and dementia has been emphasized. Cerebral small vessel disease is a common neurocognitive disorder and source of disability. Pathophysiology of cSVD is complex, involving multiple pathways, as described before. Several risk factors, including genetic, co-morbid complications, and environmental factors, contribute to the pathogenesis or exacerbate the complications. Inflammation, chronic hypoperfusion, oxidative damage, glymphatic alterations, and BBB disruption might be potential contributors to the pathogenesis of this complex phenomenon. MMPs, ROS, and other reactive factors trigger inflammatory responses, leading to the abnormalities in small vessels and endothelium dysfunction associated with CSVD.

This study has several limits: Although comprehensive, the approach used in the examined investigations in the attempt to disentangle the complex pathomechanisms of VCI has a number of caveats and potential criticisms. So far in our study, we have tried to have the most homogenous definition, but otherwise, just examining different animal models could represent not a constant level of clinical reversibility. Therefore, the available results on a relatively small sample size might not be confirmed on larger populations, although most of them were obtained from homogeneous samples.

Another limitation is that the correlation between different techniques and the anatomical distribution and severity of vascular lesions has been rarely systematically investigated; therefore, without the contribution of advanced imaging, blood samples, cerebrospinal fluid, laboratory models, or the combination of techniques, the conclusions that can be reached cannot be sufficiently powerful.

Finally, results do not usually provide specific clinical information, although they are sensitive to the “global weight” of several biochemical pathways and neurotransmitter activities. Consequently, a panel of changes, rather than a single marker of disease, should be considered.

More detailed investigations are required to understand the pathophysiology of SVD. Several fluid biomarkers that might be used in diagnostic settings have been identified. Thus, currently, there is little value in blood tests. CSF biomarkers may help physicians separate vascular and neurodegenerative causes based on BBB disruption and extracellular matrix breakdown. Alongside the need for a correct diagnosis of the disease, biomarkers could be valuable tools to monitor the progression of the disease itself and the possible response to treatment. In this work, we have tried to underline the importance of the inflammatory response in disease pathogenesis. Much further work needs to be conducted along with these positions. The search for an optimal panel of biomarkers with high sensitivity and specificity will provide the crucial tools to enhance success in identifying valid biomarkers in SVD. A combination of biochemical and imaging markers and psychometrics will be necessary to improve the diagnostic accuracy progression of the pathology and finally to monitor response to possible treatment. We believe that the contribution of inflammation on SVD is significant and should be further studied to identify new therapeutic possibilities.

## Data Availability

Not applicable.

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
