# Peer review of "Small Vessel Disease: Ancient Description, Novel Biomarkers"

_ijms, 2022, doi:10.3390/ijms23073508_

Round 1
Reviewer 1 Report
The authors deal with a fascinating timely topic, i.e., the cerebral small vessel disease (SVD), which is one of the most frequent pathological conditions leading to dementia. Indeed, although microvascular alterations causing SVD have common origins, similar cognitive outcomes, and conventional cardiovascular risk factors, the arteriolosclerosis process which underlines SVD development is based on different mechanisms, not all completely understood. These include chronic hypoperfusion state and neuroinflammation, that eventually induce endothelium activation and tissue remodeling. In this paper, the authors addressed the contribution of the chronic neuroinflammatory condition due to SVD and subsequently verified the possible focus of new biomarkers, endothelial dysfunction, and oxidative damage, which could be future targets of focused therapies. Overall, the review is nicely conceived and well written; the studies included are relevant and their main findings adequately illustrated and discussed. I have few comments only to the authors.
Abstract: this section may be summarized in order to better focus on the target of the review and report the main findings from the studies reviewed. Please also rephrase the term “slight vessel dementia”; the same holds true for “historical cardiovascular risk factors” (i.e., “conventional cardiovascular risk factors”).
Introduction: the acronym VCI (vascular cognitive impairment) is included in the abbreviation list but it is not used throughout the text. More importantly, I think that the basic concept and neurobiology of VCI should be shortly introduced.
Additional pathophysiological insights have recent come from transcranial Doppler studies, which provides useful indices of the occurrence and severity of small vessel disease in VCI patients at risk of future dementia, including those with vascular depression.
Among the potential therapeutic strategies, the role of phenolic acids (and in particular the polyphenols) and other neuroprotective agents in the prevention or management of cognitive decline (including VCI) needs to be briefly mentioned (as recently reviewed), as well as the positive impact of some dietary compounds (such as caffeine) on cognitive and mood status of VCI patients. Finally, among non-pharmacological approaches, neuromodulatory interventions (such as those based on repetitive TMS) can be proposed in VCI subjects.
Regarding the role of mitochondrial dysfunction and oxidative stress in both neurodegenerative and vascular diseases, please also highlight that even mitochondrial DNA mutation may lead to early-onset subcortical ischemic vascular dementia in adults, as previously reported.
Conclusions: please briefly list the main limitations of the studies reviewed, along with possible solutions.
Author Response
COMMENTS OF THE REVIEWER 1:
The authors deal with a fascinating timely topic, i.e., cerebral small vessel disease (SVD), which is one of the most frequent pathological conditions leading to dementia. Indeed, although microvascular alterations causing SVD have common origins, similar cognitive outcomes, and conventional cardiovascular risk factors, the arteriolosclerosis process which underlines SVD development is based on different mechanisms, not all completely understood. These include chronic hypoperfusion state and neuroinflammation, that eventually induce endothelium activation and tissue remodeling. In this paper, the authors addressed the contribution of the chronic neuroinflammatory condition due to SVD and subsequently verified the possible focus of new biomarkers, endothelial dysfunction, and oxidative damage, which could be future targets of focused therapies. Overall, the review is nicely conceived and well written; the studies included are relevant and their main findings are adequately illustrated and discussed. I have a few comments only to the authors.
- Abstract: this section may be summarized in order to better focus on the target of the review and report the main findings from the studies reviewed. Please also rephrase the term “slight vessel dementia”; the same holds true for “historical cardiovascular risk factors” (i.e., “conventional cardiovascular risk factors”).
- Introduction: the acronym VCI (vascular cognitive impairment) is included in the abbreviation list but it is not used throughout the text. More importantly, I think that the basic concept and neurobiology of VCI should be shortly introduced.
- Additional pathophysiological insights have recent come from transcranial Doppler studies, which provides useful indices of the occurrence and severity of small vessel disease in VCI patients at risk of future dementia, including those with vascular depression.
- Among the potential therapeutic strategies, the role of phenolic acids (and in particular the polyphenols) and other neuroprotective agents in the prevention or management of cognitive decline (including VCI) needs to be briefly mentioned (as recently reviewed), as well as the positive impact of some dietary compounds (such as caffeine) on cognitive and mood status of VCI patients. Finally, among non-pharmacological approaches, neuromodulatory interventions (such as those based on repetitive TMS) can be proposed in VCI subjects.
- Regarding the role of mitochondrial dysfunction and oxidative stress in both neurodegenerative and vascular diseases, please also highlight that even mitochondrial DNA mutation may lead to early-onset subcortical ischemic vascular dementia in adults, as previously reported.
- Conclusions: please briefly list the main limitations of the studies reviewed, along with possible solutions.
Dear Sir,
Thank you for your efforts to ameliorate our works, your competence, and your purposeful comments.
- We have rewritten it, highlighted in yellow.
- We have inserted some parts on VCI, lines 81-90. Highlighted in yellow
- We have inserted a brief description of transcranial Doppler, lines 91-100, highlighted in yellow.
- We have inserted a new paragraph, number 6, with a brief description of potential and previous therapies, following your suggestion, and some lines have been dedicated to TMS, too, line 393-427 (highlighted in yellow)
- Underlined this point and inserted in yellow, line 241-250
- We have inserted some of the many limits of this study.
Thank you again for your help.
Kindest regards
Rita Moretti
Reviewer 2 Report
This is a very comprehensive review on the small vessel disease (SVD) as one of the major players behind vascular dementia. Authors go on to discuss microvascular and chronic brain inflammation and endothelial dysfunction and tissue remodeling as the pathophysiological culprit in small vessel dementia. The core of this dense paper is the presentation of biomarkers that reflect endothelial dysfunction and oxidative damage. For these compounds such as endothelium-derived nitric oxide-vasodilators (ENOV), prostacyclin, CRP, and endothelium-derived hyperpolarizing factors were reviewed. Furthermore, hypoxia-endothelial-related markers were reviewed. Finally, the authors conclude that biochemical and neuroimaging markers might help correctly identify the clinical diagnosis of such brain disease.
The review is generally well-written and well-illustrated and I have only minor suggestions, as follows:
- Authors should highlight the most important biomarkers in a way that they are presented in the table and their main functions with regards to SVD explained in the respective column.
- A graphical summarizing figure highlighting the main points in this review should be provided.
- Some information on contemporary imaging methods with regard to SVD and chronic brain lesions should be expanded in the text.
Author Response
This is a very comprehensive review on the small vessel disease (SVD) as one of the major players behind vascular dementia. Authors go on to discuss microvascular and chronic brain inflammation and endothelial dysfunction and tissue remodeling as the pathophysiological culprit in small vessel dementia. The core of this dense paper is the presentation of biomarkers that reflect endothelial dysfunction and oxidative damage. For these compounds such as endothelium-derived nitric oxide-vasodilators (ENOV), prostacyclin, CRP, and endothelium-derived hyperpolarizing factors were reviewed. Furthermore, hypoxia-endothelial-related markers were reviewed. Finally, the authors conclude that biochemical and neuroimaging markers might help correctly identify the clinical diagnosis of such brain disease.
The review is generally well-written and well-illustrated and I have only minor suggestions, as follows:
- Authors should highlight the most important biomarkers in a way that they are presented in the table and their main functions with regards to SVD explained in the respective column.
- A graphical summarizing figure highlighting the main points in this review should be provided.
- Some information on contemporary imaging methods with regard to SVD and chronic brain lesions should be expanded in the text.
Dear Sir,
Thank you for your comments and your work.
- We have added a row to demonstrate point-by-point the biomarker's role in SVD, as suggested (table has been highlighted in blue)
2. We have added a graphic that might help to summarize the entire work
3. We have added the neuroimaging scale and diagnostic criteria from lines 73-81, highlighted in blue
Thank you again
Rita Moretti
This manuscript is a resubmission of an earlier submission. The following is a list of the peer review reports and author responses from that submission.